# Influence of Tool Length and Profile Errors on the Inaccuracy of Cubic-Machining Test Results

**Zongze Li** [1], **Hiroki Ogata** [1], **Ryuta Sato** [1,*], **Keiichi Shirase** [1] and **Shigehiko Sakamoto** [2]

1 Department of Mechanical Engineering, Kobe University, 1-1 Rokko-dai, Nada, Kobe 657-8501, Japan; 175t378t@stu.kobe-u.ac.jp (Z.L.); 211t311t@stu.kobe-u.ac.jp (H.O.); shirase@mech.kobe-u.ac.jp (K.S.)
2 Department of Mechanical Engineering, Kanazawa Institute of Technology, 3-1 Yatsukaho, Hakusan 924-0838, Japan; sak@neptune.kanazawa-it.ac.jp
* Correspondence: sato@mech.kobe-u.ac.jp; Tel.: +81-78-803-6326

**Abstract:** A cubic-machining test has been proposed to evaluate the geometric errors of rotary axes in five-axis machine tools using a $3 \times 3$ zone area in the same plane with different tool postures. However, as only the height deviation among the machining zones is detected by evaluating the test results, the machining test results are expected to be affected by some error parameters of tool sides, such as tool length and profile errors, and there is no research investigation on how the tool side error influences the cubic-machining test accuracy. In this study, machining inaccuracies caused by tool length and tool profile errors were investigated. The machining error caused by tool length error was formulated, and an intentional tool length error was introduced in the simulations and actual machining tests. As a result, the formulated and simulated influence of tool length error agreed with the actual machining results. Moreover, it was confirmed that the difference between the simulation result and the actual machining result can be explained by the influence of the tool profile error. This indicates that the accuracy of the cubic-machining test is directly affected by tool side errors.

**Keywords:** five-axis machine tools; cubic-machining test; tool length error; tool profile accuracy

## 1. Introduction

Five-axis machine tools have been increasingly applied in free-formed surface machining because they can control the positional displacement and the relative orientation between the cutter and the workpiece. Five-axis machine tool technology plays a crucial role in advanced manufacturing [1,2]. Although additive manufacturing has been introduced and is used for its ease of use in creating a free-form workpiece [3,4], its major limitation is that the mechanical strength of the workpiece cannot satisfy the requirements in actual cases, due to which it is impossible to replace five-axis machining with other technologies. However, compared with conventional three-axis machine tools, five-axis machine tools present more error sources, as they are composed of three translational axes and two rotary axes; thus, high-precision manufacturing requirements are difficult to satisfy. Hence, it is necessary to evaluate the error sources in five-axis machine tools.

Among all error sources, geometric errors represent the largest proportion (more than 30%) and influence the machining accuracy and directly affect the machining performance of a certain five-axis machine tool [5,6]. Non-machining- and machining-based methods can be applied for geometric error identification. The non-machining method evaluates the geometric error referred to as a professional instrument, such as a ball bar or an R-test. In ISO 10791-6 [7], geometric error identification has been defined in detail. Chen et al. [8] developed a novel method to precisely identify the geometric error through a touch-trigger probe and a sphere.

However, machine tool manufacturers and customers prefer an easy-understanding method to detect the accuracy of the machine tool rather than the use of professional instruments. Thus, the machining-based method has been considerably applied for geometric

error identification. In this method, geometric errors are detected by machining a standard machining specimen and evaluating the results. In ISO 10791-7 [9], there are some types of standard machining specimens, such as the cone frustum test and S-shaped test. However, many studies have suggested that although the S-shaped test can evaluate the integrated accuracy of the machine tool, it does not evaluate the geometric errors individually [10–14].

A cubic-machining test was proposed in the industrial field to evaluate five-axis machine tools [15], and its application has been investigated [16]. For the cubic-machining test, the evaluation or geometric error arises from the height deviation among the machining zones. However, the height deviation is also significantly affected by tool errors, which leads to a disturbance in the accuracy of the machine tool evaluation. It has been recognized that tool inaccuracy, such as tool length errors and tool profile errors, would contribute significantly to machining imprecision and should be predicted and compensated academically. Yang et al. [17] compensated for the tool length deformation caused by thermal errors and achieved an accuracy prediction of 94%. Although it is necessary to investigate the influence of tool side errors to make correct evaluations, there has been no research work on this issue until now.

The purpose of this study is to investigate the influence of the tool side error parameters on the cubic-machining test. In this study, the tool length and tool profile errors were considered. The tool length error was formulated to suggest the height deviation among the machining zones in the cubic-machining test. Both simulation and actual machining experiments were implemented in this study, with an intentional tool length error. In addition, the profile of the ball-end mill cutter was measured. The results verify that the accuracy of the cubic-machining test is directly affected by the error parameters on the tool side.

## 2. Cubic-Machining Test

A vertical-type five-axis machining center with tilting and rotary axes on the table side (NMV 1500 DCG, DMG Mori) was used in this study, in which a tilting rotary table was controlled around the B-axis and C-axis motions, as shown in Figure 1. In a previous study, the authors conducted a simulation and experiment focusing on the geometric error influences of the cubic-machining test [16].

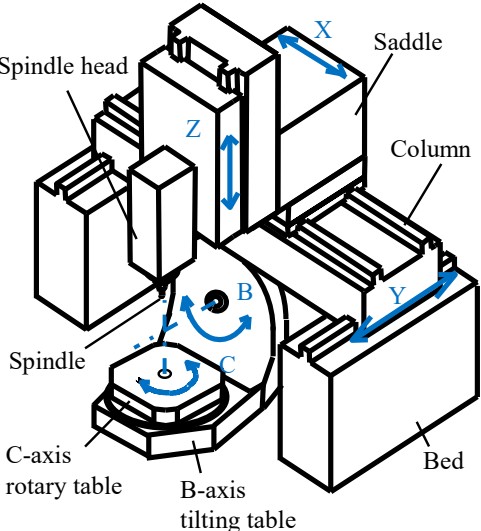

**Figure 1.** Structural configuration of a five-axis machining center.

To avoid the influence induced by other error sources, the workpiece design and machining tool path were assumed to be the same as those in [16]. Hence, as shown in Figure 2, the size of the machining test specimen was 48 × 48 mm, and the tool path of each machining zone was in the zigzag direction, and the scanning path interval was set to

0.1 mm. In addition, the machining zones can be classified into three types as shown in Figure 2: the center of the square area is ZONE I, where the tool is always vertical to the machining surface; the four squares adjacent to the center zone compose ZONE II, where the tool is tilted 30° to the normal line of the workpiece surface, toward the center of the square area; and the four squares diagonal to the center zone are ZONE III, where the tool is tilted by 30° and rotated by 45° toward the center of the square.

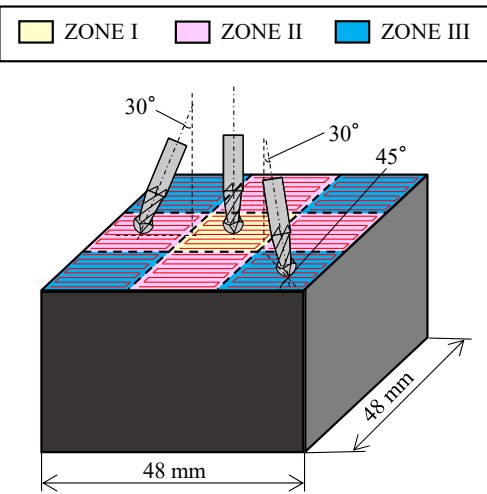

**Figure 2.** Cubic-machining test.

In the actual machining tests, the applied material of the workpiece was aluminum 7075, and the type of cutter was DLC2MBR0300, a ball-end mill produced by Mitsubishi Materials. Table 1 lists the details of the cutter. During the machining process, the finishing cutting depth was set to 0.1 mm, and the feed rate was 2000 mm/min with a spindle speed of 6000 rpm. In addition, a zero-depth cutting process was adopted after the first cutting to remove the imprecision caused by tool deflection due to cutting force.

**Table 1.** Tool specification.

| Tool Type | Ball-End Mill, DLC2MBR0300 from Mitsubishi Materials |
| --- | --- |
| Tool diameter | $\phi$ 6 mm |
| Number of flutes | 2 |
| Helix angle | 30° |
| Coating | DLC |

Figure 3a shows the actual machining experiment, and Figure 3b shows the cubic-machining workpiece after finishing cutting. Surface measurements were performed using a contact-type shape measurement system (DSF900, Kosaka Laboratory Ltd.), as shown in Figure 4a, and a $40 \times 40$ mm$^2$ square area of the machined surface was measured to evaluate the height deviation among the machining zones, as shown in Figure 4b.

In this study, as the evaluation standard represented the height deviation among machining zones, the measured surface property was assumed to be the profile of the square area surface. The surface profile of the square area was generated using a series of scanning paths of the stylus; the position deviation of the stylus was recorded to calculate the profile of the scanning path. Within a scanning measurement path, the sampling frequency was 10 mm$^{-1}$, and the interval between the scanning paths was 2 mm. While generating the surface profile, the interpolation among the scanning profiles was linear. Moreover, the average height of each zone was calculated using the generated surface profile.

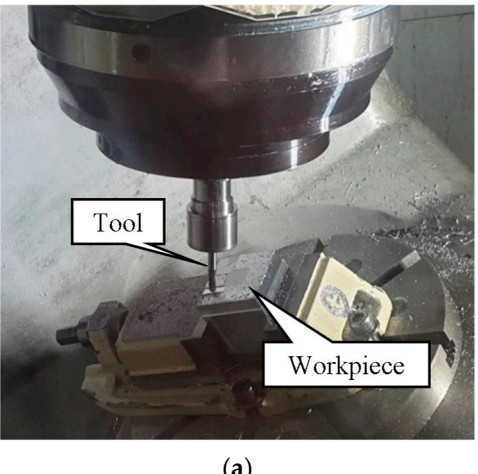
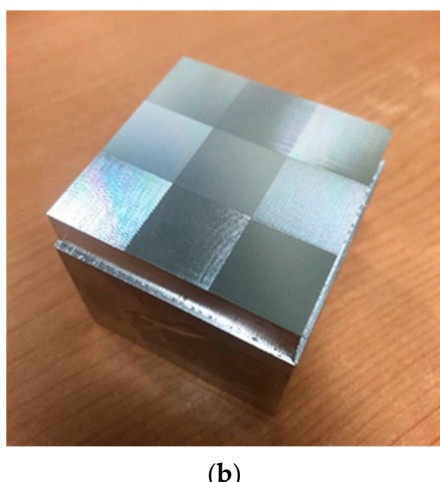

(**a**)                                    (**b**)

**Figure 3.** Actual cubic-machining test: (**a**) workpiece setting and machining method; (**b**) example of a machined workpiece.

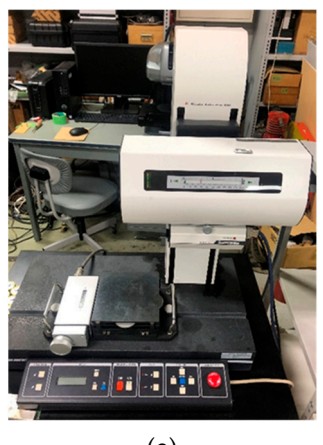
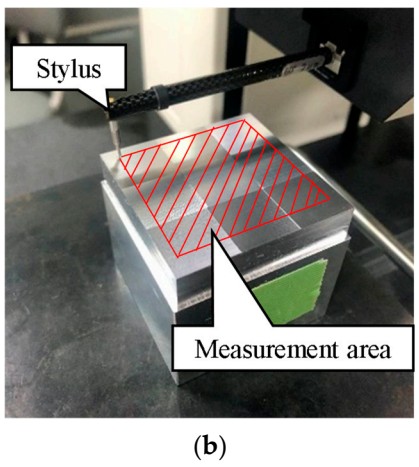

(**a**)                                    (**b**)

**Figure 4.** Measurement of machined workpiece accuracy: (**a**) measurement system; (**b**) measurement area.

## 3. Formulation of Tool Length Error Influence

Tool length error during the machining test can easily occur because of tool length measurement error, repeatability of tool change, and thermal expansion of the spindle. In this section, the influence of the tool length error on the machined accuracy is formulated to suggest the expected results. Figure 5 illustrates the height deviation caused by the tool length error between ZONE I and ZONES II and III. According to the definition of the cubic-machining test, the height deviation between ZONE I and other machining zones is assumed to be zero in an ideal situation. However, if the actual tool length is longer than the measured tool length and the deviation amount is $e_L$, the cutting depth should be larger than the ideal cutting depth. In the case of ZONE I, the tool is perpendicular to the machined surface, and the deviation in the cutting depth is also assumed to be $e_L$. In the case of ZONES II and III, as the tool has a relative angle $\theta$ to the machined surface, the deviation in the cutting depth, $e_L'$, is not supposed to be equal to $e_L$, as shown in Figure 5. The relationship between $e_L$ and $e_L'$ is calculated using Equation (1).

$$e_L' = e_L \cos \theta \tag{1}$$

Hence, the height deviation $\Delta h$ among ZONE I and other machining zones caused by the tool length error can be formulated as Equation (2).

$$\Delta h = e_L - e_L' = e_L(1 - \cos \theta) \tag{2}$$

The final machining error was attributed to numerous factors apart from tool length errors; thus, the error value $e_L'$ cannot reflect the actual machining error, and Equations (1) and (2) were applied separately to formulate the influence of the tool length error. Moreover, the formulated height deviation $\Delta h$ was applied to determine inaccuracy on the machined surface due to tool length errors existing in a certain machining.

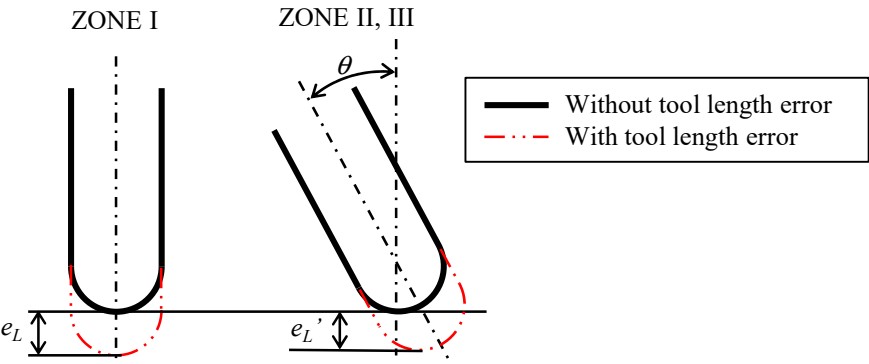

**Figure 5.** Illustration of tool length error influence.

## 4. Simulation and Experiment with Tool Length Error

### 4.1. Tool Length Measurement

In the case of actual machining, as the spindle runs for a long time, some thermal deformation occurs in the spindle, which leads to tool length instability [18]. To avoid the measurement imprecision of the tool length from spindle thermal deformation, the tool length measurement should be implemented under thermally stable conditions.

The thermal stable time region can be determined by monitoring the tool length deviation with a long-term spindle rotation. In this study, we designed three sets of experiments to monitor the tool length deviation, and the experimental conditions are listed in Table 2. The objective of these experiments was to clarify the thermal stability time of the spindle rotation, which would contribute to the inaccuracy of the tool length value; accordingly, all experiments were operated with spindle idling rotation, without any cutting load or feed motions.

**Table 2.** Experiment conditions of tool length measurement.

|  | Experiment I | Experiment II | Experiment III |
|---|---|---|---|
| Measurement time interval (min) | 5 | 30 | 30 |
| Total time (min) | 70 | 180 | 390 |

Figure 6 shows the measured results of the tool length for all three sets of experiments. During the experimental process, the tool and tool holder were kept fixed to the spindle, such that the error due to the tool changing accuracy could be assumed to be 0. The experimental value shown in Figure 6 is the change in the tool length from the value measured at the initial time of each experiment. According to Figure 6, after approximately 120 min of spindle running, the deviation in the tool length became stable. Thus, as the thermal deformation of the spindle stabilizes after 120 min, the tool length measurement and machining tests should be implemented after 120 min of spindle rotation.

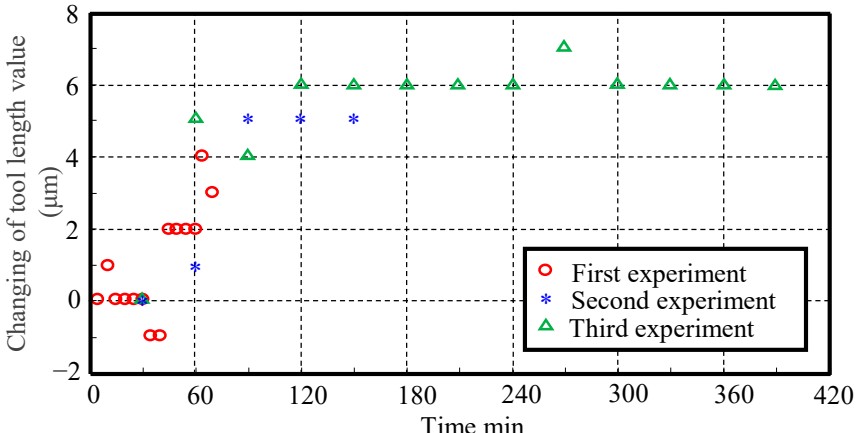

**Figure 6.** Measured thermal expansion characteristics on the tool length changes.

### 4.2. Experiment with Tool Length Errors

In this study, the influence of the tool length error of the cubic-machining test and the validity of Equation (2) in actual machining were verified. For the machining tests, the tool center point (TCP) control mode with a tool length setting was adopted. This means that the controller controlled the position of the axes based on the tool length parameter. Therefore, the tool length $e_L$ was intentionally set to a certain value added to the measured tool length and set to the controller. The given values of $e_L$ were 0 and $\pm 10$ μm. In addition, to remove the tool length deviation caused during the machining process, the authors measured the tool length individually before machining each zone. It was also confirmed that the change in tool length was approximately 1 μm.

Figure 7 shows the measured machined surface of the cubic machining with different values of tool length error $e_L$, where the values (unit: mm) attached to each machining zone are the average relative height deviation from the standard machining zone (ZONE I), and the deviation appearing on the machining without tool length error ($e_L = 0$) was considered to be caused by geometric errors of the machine tool. According to Figure 7, it can be seen that the average height of the surfaces became higher when the negative tool length error existed.

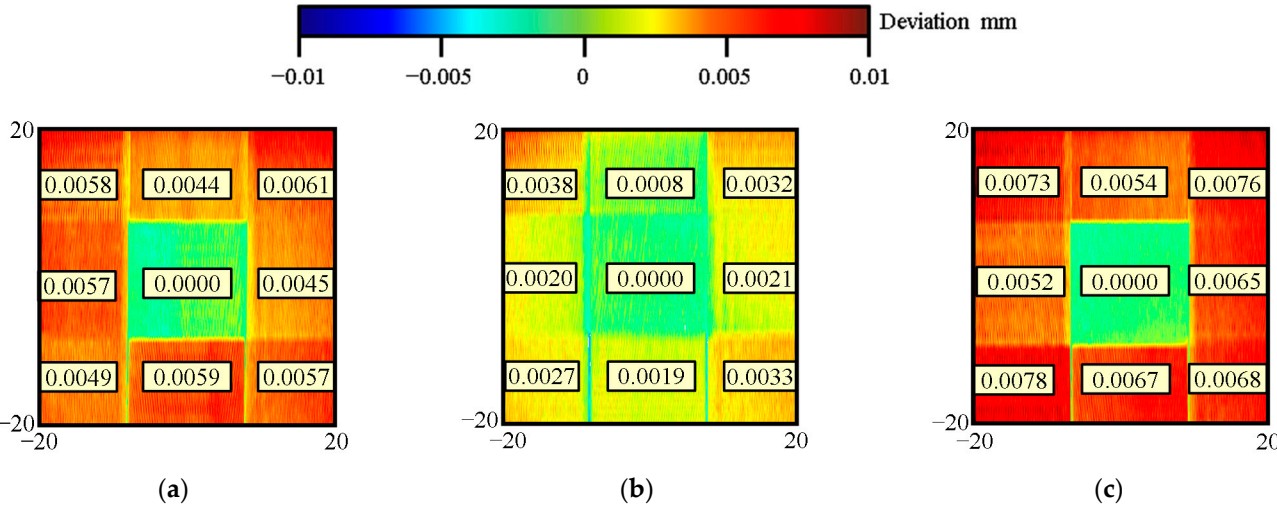

**Figure 7.** Measured relative height deviation of the test surfaces: (**a**) without tool length error; (**b**) with positive tool length error; (**c**) with negative tool length error.

The influence of tool length error can be evaluated by considering the difference between the average heights of the surfaces with and without tool length errors, because

the influence of other factors, such as geometric errors, can be assumed to be constant. Hence, the influence or tool length error can be qualified based on the differences between homologous zones with and without tool length errors, as shown in Figure 7. Table 3 shows the relative average height of the surfaces compared with the results without tool length errors. As the tool length error values were set to ±10 µm, according to Equation (2), $\Delta h$ caused by tool length error was assumed to be ±1.3 µm. However, according to Table 3, there are 1–3 µm differences between the deviations that appear in the actual machining test. The influence of the tool length error on the actual machined accuracy did not agree with the theoretical value. Thus, other factors, such as the tool path for machining, may have affected the results, which will be further evaluated. Nonetheless, it was confirmed that the tool length error directly affected the machined accuracy, and the tendency of the influence of the error was in accordance with the expected.

**Table 3.** Average height differences between zones with and without tool length error (experiment).

| Machining Zones | Positive Tool Length Error Case (µm) | Negative Tool Length Error Case (µm) |
|---|---|---|
| ZONE II-1 | −3.7 | −0.5 |
| ZONE II-2 | −4.0 | 0.8 |
| ZONE II-3 | −2.4 | 2.0 |
| ZONE II-4 | −3.6 | 1.0 |
| ZONE III-1 | −2.2 | 2.9 |
| ZONE III-2 | −2.4 | 1.1 |
| ZONE III-3 | −2.9 | 1.5 |
| ZONE III-4 | −2.0 | 1.5 |

*4.3. Simulation with Tool Length Error*

To clarify the influence of the tool length error, the machined accuracy was simulated considering the geometric and tool length errors. It has been confirmed that the positional and angular commands of each axis can also cause machining inaccuracies [19]. Therefore, simulations were performed based on the calculated position and angle of each axis obtained from the Numerical Control (NC) program used for the actual machining tests. Consequently, it was established that the simulation results were only influenced by geometric and tool length errors.

The simulation process used is shown in Figure 8. The positions of the X-, Y-, and Z-axes represent the translational feed motion of the spindle, and those of the B- and C-axes indicate the orientation of the work table. According to [14], the coordinate transformation from the machine coordinate system to the workpiece coordinate transformation was implemented using Equations (3) and (4), respectively:

$$P_{M,t} = [X, Y, Z - tl, 1]^T \tag{3}$$

$$P_{W,t} = M_{WC} \cdot M_c \cdot M_{\alpha CB} \cdot M_{\delta xCB} \cdot M_B \cdot M_{\gamma BY} \cdot M_{\beta BY} \cdot M_{\alpha BY} \cdot M_{\delta zBY} \cdot M_{\delta yBY} \cdot M_{\delta xBY} \cdot P_{M,t} \tag{4}$$

where X, Y, and Z are the positions of the X-, Y-, and Z-axes, respectively; *tl* is the tool length; $P_{M,t}$ and $P_{W,t}$ are the homogeneous coordinates of the tool tip points under the machine and workpiece coordinate systems, respectively; and $M_C$ and $M_B$ represent the feed motion of the C-axis and B-axis, respectively, which are described through the D–H matrix as follows:

$$M_B = \begin{bmatrix} \cos B & 0 & \sin B & 0 \\ 0 & 1 & 0 & 0 \\ -\sin B & 0 & \cos B & 0 \\ 0 & 0 & 0 & 1 \end{bmatrix}, \ M_C = \begin{bmatrix} \cos C & -\sin C & 0 & 0 \\ \sin C & \cos C & 0 & 0 \\ 0 & 0 & 1 & 0 \\ 0 & 0 & 0 & 1 \end{bmatrix} \tag{5}$$

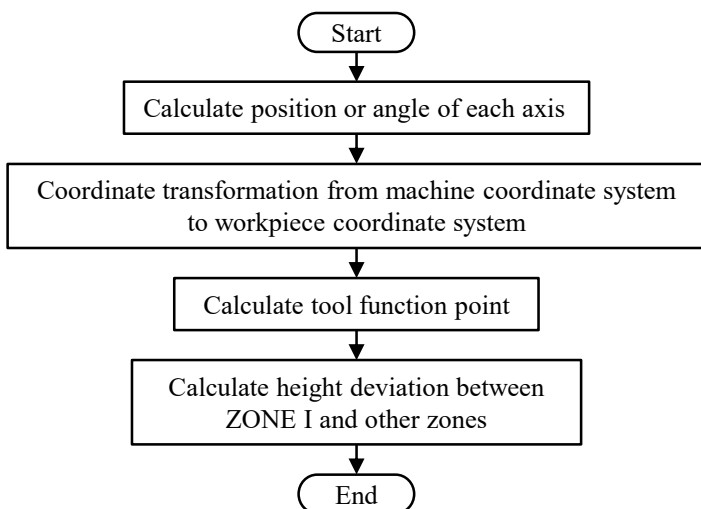

**Figure 8.** Simulation process.

In addition, $M_{\gamma BY}$, $M_{\beta BY}$, $M_{\alpha BY}$, $M_{\delta zBY}$, $M_{\delta yBY}$, and $M_{\delta xBY}$ are the impact matrices of geometric errors between the B-axis and machine bed, and $M_{\alpha CB}$ and $M_{\delta xCB}$ are the impact matrices of geometric errors between the B- and C-axis, respectively. The definitions of each geometric error are presented in Table 4 and illustrated in Figure 9.

**Table 4.** Geometric errors in the five-axis machine tool with B- and C-axis on table side.

| Symbol | Description |
|---|---|
| $\delta_{xBY}$ | Positional error of B-axis average line along X-axis direction |
| $\delta_{yBY}$ | Positional error of B-axis average line along Y-axis direction |
| $\delta_{zBY}$ | Positional error of B-axis average line along Z-axis direction |
| $\alpha_{BY}$ | Angular error between B-axis and Y-axis around X-axis direction |
| $\beta_{BY}$ | Angular error between B-axis and Y-axis around Y-axis direction |
| $\gamma_{BY}$ | Angular error between B-axis and Y-axis around Z-axis direction |
| $\alpha_{CB}$ | Angular error between C-axis and B-axis around X-axis direction |
| $\delta_{xCB}$ | Positional error of C-axis and B-axis along X-axis direction |

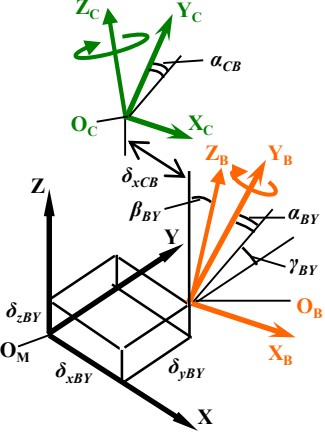

**Figure 9.** Illustration of geometric errors listed in Table 4.

Based on the description in Table 3, the impact matrix of each geometric error is defined by Equations (6)–(11). Geometric errors were identified for the simulations using a ball bar [20]. The identified geometric errors are listed in Table 5.

$$M_{\delta zBY} \cdot M_{\delta yBY} \cdot M_{\delta xBY} = \begin{bmatrix} 1 & 0 & 0 & \delta_{xBY} \\ 0 & 1 & 0 & \delta_{yBY} \\ 0 & 0 & 1 & \delta_{zBY} \\ 0 & 0 & 0 & 1 \end{bmatrix} \tag{6}$$

$$M_{\alpha BY} = \begin{bmatrix} 1 & 0 & 0 & 0 \\ 0 & \cos\alpha_{BY} & -\sin\alpha_{BY} & 0 \\ 0 & \sin\alpha_{BY} & \cos\alpha_{BY} & 0 \\ 0 & 0 & 0 & 1 \end{bmatrix} \tag{7}$$

$$M_{\beta BY} = \begin{bmatrix} \cos\beta_{BY} & 0 & \sin\beta_{BY} & 0 \\ 0 & 1 & 0 & 0 \\ -\sin\beta_{BY} & 0 & \cos\beta_{BY} & 0 \\ 0 & 0 & 0 & 1 \end{bmatrix} \tag{8}$$

$$M_{\gamma BY} = \begin{bmatrix} \cos\gamma_{BY} & -\sin\gamma_{BY} & 0 & 0 \\ \sin\gamma_{BY} & \cos\gamma_{BY} & 0 & 0 \\ 0 & 0 & 1 & 0 \\ 0 & 0 & 0 & 1 \end{bmatrix} \tag{9}$$

$$M_{\alpha CB} = \begin{bmatrix} 1 & 0 & 0 & 0 \\ 0 & \cos\alpha_{CB} & -\sin\alpha_{CB} & 0 \\ 0 & \sin\alpha_{CB} & \cos\alpha_{CB} & 0 \\ 0 & 0 & 0 & 1 \end{bmatrix} \tag{10}$$

$$M_{\delta xCB} = \begin{bmatrix} 1 & 0 & 0 & \delta_{xCB} \\ 0 & 1 & 0 & 0 \\ 0 & 0 & 1 & 0 \\ 0 & 0 & 0 & 1 \end{bmatrix} \tag{11}$$

**Table 5.** Identified geometric errors in the machine tool.

| Item | $\delta_{xBY}$ (μm) | $\delta_{yBY}$ (μm) | $\delta_{zBY}$ (μm) | $\alpha_{BY}$ (°) | $\beta_{BY}$ (°) | $\gamma_{BY}$ (°) | $\alpha_{CB}$ (°) | $\delta_{xCB}$ (μm) |
|---|---|---|---|---|---|---|---|---|
| Value | 2.0 | −11.0 | −0.647 | 0.0017 | 0.0019 | 0.0210 | −0.0054 | 9.7 |

The tool tip position, considering the geometric and tool length errors, can be calculated as mentioned above. To simulate the machined accuracy, the position of the functional point is required. The relationship between the tool tip point and tool functional point in the cubic-machining test is illustrated in Figure 10. Therefore, the tool functional point can be calculated using Equation (12), where the tool posture $v$ is calculated using Equation (13), where $r$ is the radius of the ball-end mill, and $P_{W,f}$ and $P_{W,t}$ are the coordinates of the tool functional point and tool tip point in Figure 10.

$$P_{W,f} = P_{W,t} + r \cdot (i,\ j, k-1) \tag{12}$$

$$\begin{bmatrix} i \\ j \\ k \end{bmatrix} = \begin{bmatrix} \cos C & -\sin C & 0 \\ \sin C & \cos C & 0 \\ 0 & 0 & 1 \end{bmatrix} \cdot \begin{bmatrix} \cos B & 0 & \sin B \\ 0 & 1 & 0 \\ -\sin B & 0 & \cos B \end{bmatrix} \cdot \begin{bmatrix} 0 \\ 0 \\ 1 \end{bmatrix} \tag{13}$$

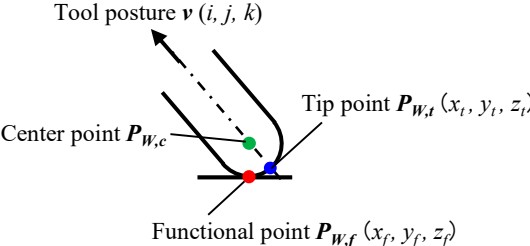

**Figure 10.** Relationship between tool center point, tool tip point, and tool functional point.

The tool functional point indicates the geometry of the simulated machined surface. Hence, the Z-axis coordinate $z_f$ expresses the height of the machine zones. The relative height deviation due to errors can be calculated using Equation (14), where $\overline{z_{f,\ ZONE\ I}}$ is the average value of $z_f$ for ZONE I, and $\overline{z_{f,ZONE\ n}}$ is the average value of $z_f$ for ZONE n (n = II-1, II-2, etc.).

$$\Delta h_n = \overline{z_{f,ZONE\ n}} - \overline{z_{f,\ ZONE\ I}} \tag{14}$$

Figure 11 shows the simulated results of cases with a different tool length error $e_L$, and the value (unit: mm) attached to the machining zones suggests the height deviation from ZONE I. According to Figure 11, the average height of the surfaces increased when a negative tool length error existed, similar to the real machined results. Table 6 shows the influences caused by tool length errors compared with the results without tool length error. According to Table 6, it can be indicated that the simulation results agree with the formulated value for each machining zone.

**Table 6.** Average height differences between zones with and without tool length error (experiment).

| Machining Zones | Positive Tool Length Error Case (μm) | Negative Tool Length Error Case (μm) |
|---|---|---|
| ZONE II-1 | −1.3 | 1.3 |
| ZONE II-2 | −1.3 | 1.4 |
| ZONE II-3 | −1.3 | 1.3 |
| ZONE II-4 | −1.3 | 1.4 |
| ZONE III-1 | −1.3 | 1.4 |
| ZONE III-2 | −1.3 | 1.4 |
| ZONE III-3 | −1.3 | 1.3 |
| ZONE III-4 | −1.3 | 1.3 |

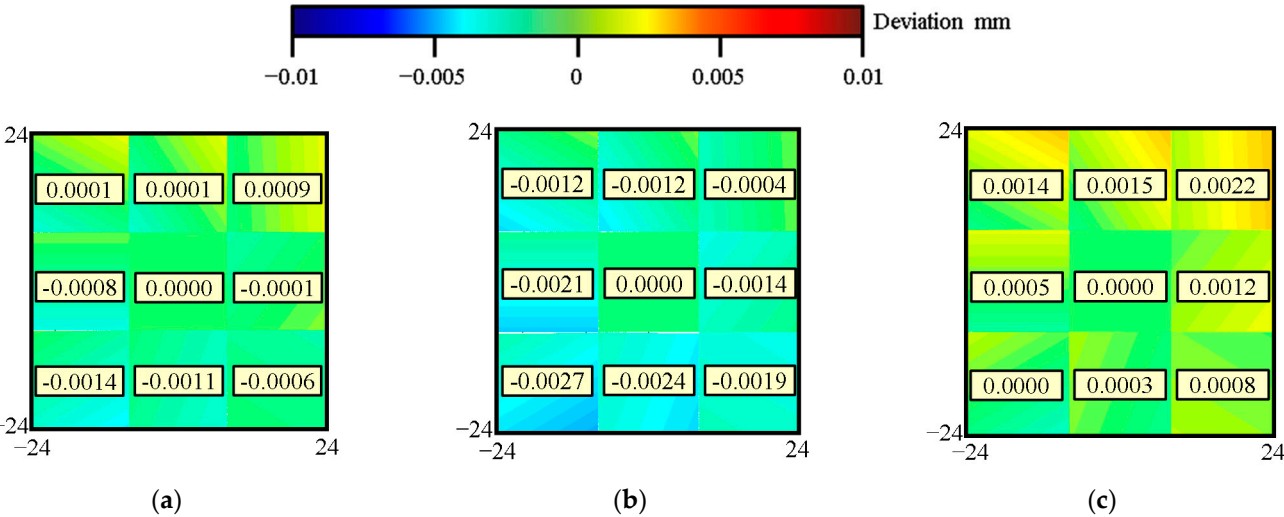

**Figure 11.** Simulated relative height deviation of the test surfaces: (**a**) without tool length error; (**b**) with positive tool length error; (**c**) with negative tool length error.

## 5. Tool Profile Accuracy Influence

Comparing the measured surfaces shown in Figure 7 and the simulated ones shown in Figure 11, the average height of the simulated surfaces was lower than that of the measured ones around 5 µm. To clarify the reason, the tool profile accuracy of the ball-end mill should be considered.

In this study, the tool profile measurement was implemented using a Dyna vision system (produced by Big Daishowa Group). Figure 12a,b shows the measurement setup and measured vision, respectively. A processor was installed inside the Dyna vision system to calculate the deviation from the profile to the standard scale. The angular interval of the measurement process was selected to be 1°, and the profile was measured until 45° from the tool tip point.

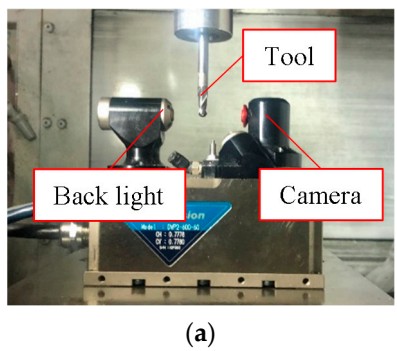

(a)

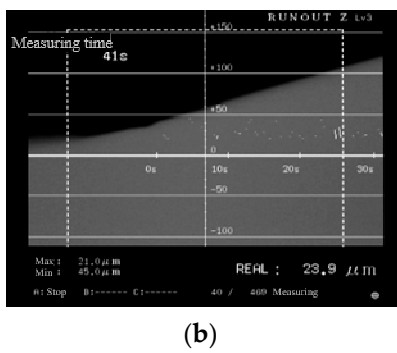

(b)

**Figure 12.** Tool profile measurement method: (**a**) measurement setup; (**b**) example of a graphical vision for the measurement.

Figure 13 shows the measured result of the tool profile accuracy, where the vertical axis represents the radial error of the tool profile, calculated as the deviation between the measured and nominal tool radii. A negative deviation indicates that the tool diameter is smaller than the designed one. In the actual cubic-machining test, ZONE I was machined by an area of approximately 0°, and the others were machined by an area of approximately 30°. According to Figure 13, there was a 4 µm deviation between the 0° and 30° areas, which is in agreement with the difference between the simulation and machining results discussed in Section 4. This suggests that the tool profile error directly affects the accuracy of the cubic-machining test. It can be said from the result that the tool profile should be carefully considered in the evaluation of the cubic-machining test accuracy. More tests will be conducted in the future with different tool accuracies to clarify the influence of tool profile errors.

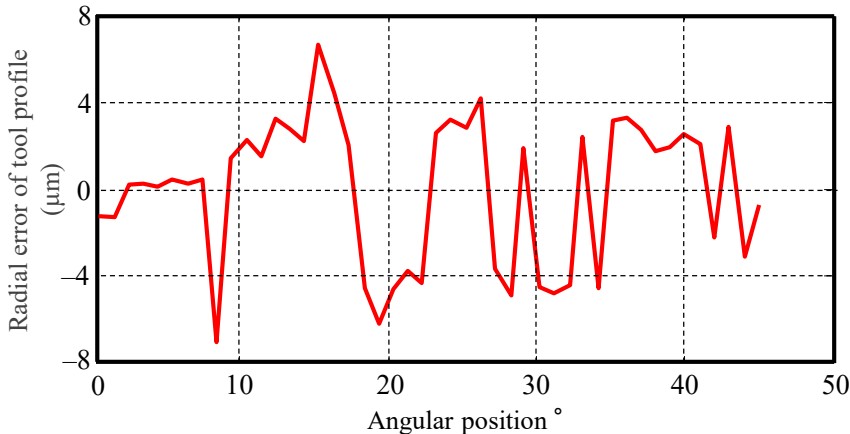

**Figure 13.** Measured tool profile accuracy of the ball-end mill used in this study.

## 6. Conclusions

In this study, the influence of tool length and tool profile errors on the accuracy of the cubic-machining test was investigated. The height deviation caused by tool length error was formulated. In addition, both the actual machining experiment and the machining simulation with tool length errors were implemented. To clarify the reason for the difference between the experimental and simulation results, the tool profile accuracy was measured, and the influence of the tool profile was analyzed. The conclusions of this study can be summarized as follows.

(1) Results of actual machining tests and simulations with intentional tool length errors mentioned in Sections 4.2 and 4.3 indicate that the tool length error directly affects the machined accuracy, and the tendency of the influence of the error is in agreement with the expected one in general.

(2) Results obtained from comparison between the deviation results in Section 4.2 and the tool profile accuracy in Section 5 suggest that the tool profile should be carefully considered in the evaluation of the cubic-machining test accuracy.

The investigation in this study indicates the direction of the evaluation of the cubic-machining test. In the future, we will focus on other error sources to clarify the residual deviation between the simulation and experimental results.

**Author Contributions:** Z.L.: writing—original draft presentation, R.S.: writing—review and editing, Z.L. and H.O.: experiments and simulations, R.S.: supervision, project administration, and funding acquisition, K.S.: laboratory supervision, and S.S.: test piece design. All authors have read and agreed to the published version of the manuscript.

**Funding:** This research was funded by the Research Project RU-10 of the Machine Tool Engineering Foundation.

**Data Availability Statement:** The data presented in this study are not available.

**Acknowledgments:** The authors would like to sincerely acknowledge all the support from DMG Mori Seiki Co., Ltd., and MTTRF (Machine Tool Technologies Research Foundation).

**Conflicts of Interest:** The authors declare no conflict of interest. The funders had no role in the study design, collection, analyses, or interpretation of data; in the writing of the manuscript; or in the decision to publish the results.

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
