# Peer review of "Influence of Tool Length and Profile Errors on the Inaccuracy of Cubic-Machining Test Results"

_jmmp, doi:10.3390/jmmp5020051_

Round 1

Reviewer 1 Report

The topic of the paper is very interesting from the theoretical and practical aspect. The applied test method (cubic machining test) is practical and simply method in order to investigate the accuracy of 5 axis milling. The title of the paper is expressive, the abstract and the list of keywords are adequate.

Questions and  comments:

  1. What was the material of the test part?
  2. What were the properties of the cutting tool? (diameter, number of teeth, material, coating…)
  3. What were the cutting parameters?
  4. 2 : Please put the tools to the middle of the zones in order to better understanding.
  5. Row 81-86: What was the measured surface property and wat was the measuring method?
  6. 1: The test was performed with or without cutting? In case of “whithout cutting” the increasing of the temperature is slower thanks to the smaller load. In cas of “with cutting” the tool wear cam modify the length of the tool. 120 min is longer than the general tool life.
  7. 2: How did the zero level and the value of the relative height were defined and calculated?
  8. 7 : The caption is not accurate, please modify. e.g.: “Average relative height deviation of the test surfaces; (a)…”
  9. Table 2: The caption is not accurate, please modify.
  10. 11: see Fig.7

Reviewer 2 Report

This paper studied the influence of tool length and profile errors on the Inaccuracy of Cubic-Machining Test Results.

  1. The authors should clearly describe the material information of the tool and the cubic.
  2. Additive manufacturing is a new trend to fabricate machining tools, the authors could discuss and mention the possibility of additive manufacturing of tools especially tools containing cooling channels inside. for example, [DOI: 10.1007/s00170-020-05389-5] [DOI: 10.1007/s40516-019-00092-0]

Author Response

1. The authors should clearly describe the material information of the tool and the cubic.
Thank you for your comments. We feel so sorry that we forgot to mention the test material as well as applied cutter in the original manuscript. The material is A7075, and we have added this information in the revised manuscript. The applied ball-end mill is DLC2MBR0300, produce by Mitsubishi Material. And we made another table to introduce the properties of the cutting tool in the revised manuscript.
2. Additive manufacturing is a new trend to fabricate machining tools, the authors could discuss and mention the possibility of additive manufacturing of tools especially tools containing cooling channels inside. for example, [DOI: 10.1007/s00170-020-05389-5] [DOI: 10.1007/s40516-019-00092-0]
Thank you for your comments. It is well-known that additive manufacturing technology is applied widely for free-form workpiece and has been a new trend to fabricate machine tools. We have added this discussion in the revised Introduction.

Reviewer 3 Report

The article titled ”Influence of tool length and profile Errors on the Inaccuracy of Cubic-Machining test Results” presents the interesting problem impact changes in tool geometry on the accuracy of five-axis milling. This is an important topic especially from a practical point of view as simple tests to check machining accuracy on the shop floor are very needful.

In contrast, the article lacks a clear presentation of the research. For example, if one factor is tested, the stability of other factors should be checked. In Chapter 3, for example, the error value eL’ does not only depend on the elongation of the tool but also on the accuracy of the spherical part of the tool. This chapter would include to describe the tool in detail, along with a detailed account of the accuracy of its execution. About the accuracy of the spherical part is only presented in Chapter 5. It should be checked at the beginning of the researches and taken into account in the interpretation of the results. In addition, the work may be included more detail comments connecting the test results with causes of errors.

Details remarks:

  • The Figure 6 and Figure 13 should be better described. What is meaning “deviation”. It is only random value or may be corrected during the process.
  • Under formulae 3, 4, 12, 14 there should be an explanation of the symbols used.
  • Conclusions contain only general statements lacking references to the research conducted.
  • DOIs are missing from the Reference.
  • Moreover, in the references, on the 6 items from 15, authors present their own work. The list of references should be changed and extended.

Reviewer 4 Report

Dear authors, 

your manuscript is well written and is presented with sufficient rigor. It would a good addition to the field.

Despite this, information about the type of materials and tools used in the experiment should be explained as the different errors could be a function of the type of machined material as friction will differ and may have an impact. I agree, this presents a more theoretical study, but the specifics of the experiment should be more clearly presented

Despite the references, additional references should be added especially since more than half of the references are yours

Author Response

your manuscript is well written and is presented with sufficient rigor. It would a good addition to the field.
Despite this, information about the type of materials and tools used in the experiment should be explained as the different errors could be a function of the type of machined material as friction will differ and may have an impact. I agree, this presents a more theoretical study, but the specifics of the experiment should be more clearly presented

Thank you for your comments. We feel so sorry that we forgot to mention the test material as well as applied cutter in the original manuscript. The material is A7075, and we have added this information at Row 92 in the revised manuscript. The applied ball-end mill is DLC2MBR0300, produce by Mitsubishi Material. And we made another table to introduce the properties of the cutting tool in the revised manuscript.

Despite the references, additional references should be added especially since more than half of the references are yours

Thank you for your comments. We feel so sorry that we did not list efficient amount of reference. The number of references has been expanded to 20 pieces.

Round 2

Reviewer 3 Report

The word "deviation" is well explained in the review response but on the Y axis in Figures 6 and 13 some adjective may be added to this word to more precisely determine this parameter.